# Running Pace Percentile Values for Brazilian Non-Professional Road Runners

**DOI:** 10.3390/healthcare9070829

**Published:** 2021-06-30

**Authors:** Mabliny Thuany, Beat Knechtle, Lee Hill, Thomas Rosemann, Thayse Natacha Gomes

**Affiliations:** 1Centre of Research, Education, Innovation and Intervention in Sport (CIFI2D), Faculty of Sports, University of Porto, 4200-450 Porto, Portugal; mablinysantos@gmail.com; 2Medbase St. Gallen Am Vadianplatz, Vadianstrasse 26, 9001 St. Gallen, Switzerland; 3Division of Gastroenterology and Nutrition, Department of Pediatrics, McMaster University, Hamilton, ON L8N 3Z5, Canada; hilll14@mcmaster.ca; 4Institute of Primary Care, University of Zurich, 8091 Zurich, Switzerland; thomas.rosemann@usz.ch; 5Department of Physical Education, Federal University of Sergipe (UFS), São Cristóvão 49100-000, Brazil; thayse_natacha@hotmail.com

**Keywords:** runners, age groups, performance

## Abstract

Background: The purpose of this study was to establish sex-specific percentile curves and values for the running pace of Brazilian non-professional runners. Methods: The sample comprised 1152 amateur runners aged 18–72 (61.8% males), from the five Brazilian regions. The runners answered an online questionnaire providing information about their biological (sex, age, height, weight) and training (volume and frequency/week, running pace) characteristics. Using 25th, 50th, and 75th percentile, the running pace was computed for women and men by age groups and by running distances (5 km, 10 km, 21 km, and 42 km). Sex- and age-specific percentile curves (10th, 25th, 50th, 75th, and 90th) were created through the Lambda Mu Sigma method. Results: For all ages and distance, men performed better than women, and a decrease in the performance was observed across age groups. Among male runners, the beginning of their thirties and the end of their forties seem to be the moments where they observed substantial improvements in running pace; among female runners, this improvement phase was observed to be more pronounced toward the end of their forties. Conclusions: Percentile values of running pace could help coaches during training programs and runners to better understand “how well” they are comparing against their peers.

## 1. Introduction

Sport performance can be defined in different ways, based on the modality considered, as well as specific abilities evaluated [1]. Namely, in road running, one of the most used indices to describe a runner’s performance is running pace [2]. Defined as the time (usually in minutes) needed to cover one kilometer (km) [2], the running pace appears to be the most popular indicator of runner’s performance and is used by coaches during prescription, control, and evaluation of training [3]. For athletes, beside the running pace as the target to be achieved during training, pacing strategy has been frequently used to control energy expenditure during running events [4].

Based on this index, and taking into account the increase in the number of road runners participating in events of the modality in least years, sports events organizers tend to split athletes into different groups (based on athletes’ reported running pace or estimated time to cover the distance) before the running starts [5,6,7,8]. Furthermore, some running events have also stablished reference values, which must be achieved by non-professional runners in order to take part in an event [9], such as the Major Marathon events (Boston, Berlin, Tokyo, London, New York, and Chicago). However, there is a lack of information about running pace reference values for Brazilian runners at different distances (from 5 km to a marathon).

Previous research has focused on establishing reference values across several different areas. For example, Miguel-Etayo et al. [10] described physical fitness reference standards for European children aged 6–10.9; reference values for fitness level and gross motor skills were also provided for Chilean children aged 4‒6 [11]; and reference values for body composition and anthropometric measurements for athletes were suggested by Santos et al. [12]. However, little is known about reference values used as indicators of sport performance for non-athletic populations.

Regarding road running, in the Brazilian context there are about 4 million non-professional runners [13]. These athletes undertake specific training sessions focusing on improving their health and quality of life [14,15,16], towards the achievement of personal goals and/or records [17,18], as well as performance, including participation in competitions in their country or even at an international level [16]. Further, the existence of national reference values related to running pace for non-elite road runners, could help runners to compare their performance amongst their peers at the same interval age, race distance, and sex. Moreover, it could help them plan more effective training sessions, in addition to providing information about the performance of non-elite Brazilian runners. Ultimately, it would allow this athletic population to be compared against international values and be an indicator of their training and performance progress. Therefore, the aim of the present study was to establish sex-specific percentile curves and values for running pace for non-elite Brazilian road runners, in the context of age, sex, and race distance. We hypothesize that for both sexes, the running pace would increase until the runners become 30 years of age, stabilizing before they are 40 years of age, followed by a new increment after this age. In addition, it is expected that male runners present better running pace values when compared against their female peers.

## 2. Materials and Methods

### 2.1. Sample

The sample of the present study comes from the InTrack project [19], a cross-sectional study aiming to identify factors associated with road running performance. For the present study, the sample comprised of 1152 runners (61.8% males), from the five Brazilian regions (North = 7.4%; Northeast = 35.7%; South = 12.4%; Southeast = 36.3%; Midwest = 8.2%). The participants gave their consent to participate in the study, and the research was conducted according to the Declaration of Helsinki. The study was approved by the Ethics Committee from the Federal University of Sergipe (protocol n° 3.558.630).

### 2.2. Procedures and Data Collection

All runners had answered the online questionnaire called “Profile description and factor associated with runner’s performance” [20] that was available for eligible participants using an online platform (Google forms) (methodology used in previous studies) [21,22,23]. The questionnaire provided information pertaining to sex, age, self-reported anthropometric variables (weight and height), sociodemographic profile, perception about the environmental influence on the training, training variables (volume and frequency/week, practice time), and family sport involvement. In the present study, the following information was extracted from the online questionnaire:

Demographic information: self-reported sex, age, height (cm), and weight (kg). The runners were split into 5-year age groups (<20 years; 20–24 years; 25–29 years; 30–34 years; 35–39 years; 40–44 years; 45–49 years; 50–54 years; 55–59 years; 60–64 years; 65–69 years; ≥70 years), based on the classification usually used in competitions carried out in Brazil [24,25]. Further, body mass index (BMI) was computed by the formula: weight (kg)/height (m^2^).

### 2.3. Training Variables

Running pace: runners were asked to state their best running pace over their preferred distance (and to indicate their preferred distance). The variables were reported as min: sec, and it was transformed in seconds, for further analysis.

Volume and frequency/week: self-reported volume/week was obtained from the questions, “on average, how many kilometres do you run per week?”; and for the training frequency, they should choose the option regarding their training frequency (1–7 units/week).

### 2.4. Statistical Analysis

Runners’ pace was tested for normality distribution, according to sex and age groups, through the Kolmogorov-Smirnov test. Descriptive statistics were presented as mean (standard deviation) and frequencies (%). Firstly, values for the 25th, 50th, and 75th percentile for running pace were computed for women and men, by age groups. In addition, the same percentiles were computed, by sex, taking into account the running distance (5 km, 10 km, 21 km, and 42 km). Chi-square test (x^2^) was computed to estimate between sexes differences, taking into account the preferred distance. These analyses were computed on SPSS 24.0, with a significant level of 5%. Taking into account information of running pace without split into running distances, sex- and age-specific percentile curves (10th, 25th, 50th, 75th, and 90th) were created through the Lambda Mu Sigma method (LMS), using the LMS chartmaker Pro (v2.54), where the best model was defined based on the goodness of fit values produced.

## 3. Results

Descriptive information is presented in Table 1. The sample was comprised for 1152 runners from both sexes (male—61.8%, *n* = 712; female—38.2%, *n* = 440), aged between 18 and 72 years, with a mean BMI of 24.2 ± 3.1 kg.m^−2^. Runners related to cover about 35.4 km/week in their training, with a frequency of three training sessions/week. Regarding the distance of preference, 58.7% of them reported prefer short distance ones, and the marathon was that one with the lowest frequency (7.5%). Comparisons by distances of preferences revealed significant differences between sexes (x^2^ = 19.2; *p* < 0.001). Women reported higher preference to short distance (5 km and 10 km), while men preferred long-distances (half-marathon and marathon). Furthermore, the mean running pace was of 324 s/km, while men showed a pace 55 s lower than women.

Results for running pace percentiles by sex and age group are presented in Table 2. Women presented a slower pace than men in all age-groups. In addition, among male runners, the best paces (expressed in the 25th percentile) are observed into the age range 19–29 years, with increments in time to cover 1 km observed in general, after this age into this percentile. On the other hand, among female runners, increments observed in pace at the 25th percentile are smaller than those observed among men.

Similarly, results from Table 3 also highlight sex differences in running pace when organized by running distance but there was no a clear trend in this variable when analyzed by distance.

Normative graphs (P10, P25, P50, P75, P90) for the running pace are represented in Figure 1 and Figure 2. For all age groups, men performed better than women, and a decrease in the performance is observed across age groups. Further, among men, at the beginning of their thirties (30–33 years of age) and at the end of their forties (47 to 49 years of age) more substantial increments in running pace were observed. Among female runners, stability in the performance seems to occur before the end of their forties, followed by an increase in running pace with increasing age.

## 4. Discussion

Running is an affordable and easily accessible sport, requiring not much expense regarding the necessity of equipment acquisition. The increase in the number of road runners and running events have been highlighted during the last two decades at an international level [26]. The present study aimed to establish sex-specific percentile curves and values for running pace for non-professional Brazilian road runners, taking into account age, sex, and race distance. Findings of the present study showed that (1) among male runners, the beginning of their thirties and the end of their forties seem to be the moments with substantial improvements in their running performance; (2) among female runners, this improvement phase was observed to be more pronounced toward the end of their forties; and (3) women presented slower running pace than men, in all age-groups.

These results can be associated with the fact that older runners tend to have more time in practice engagement, which can lead them some advantages related to the development of physiological aspects and running strategy. Furthermore, in Brazilian context, there is a higher concentration of practitioners in the age range 30–40 years [27,28,29,30], with a sub-representation of young and elderly people [31]. It has been noted, during the last years, a change in runners’ profile, with runners getting older (when considered the mean age of the groups) [32], leading to a reduction in performance. Differently from our results, the study conducted by Nikolaidis et al. [33] identified that when analyzed in the 5-years age intervals, the age peak performance occurred between 30 and 34 years old in women, and 35–39 years in men. When data were shown at 1-year age intervals, women presented the best race time at 29.7 years and men at 34.8 years. Differences in results can be associated with differences in the means of age, since women in the present study were older than those from Nikolaidis et al.’s [33], as well as due to differences in the competitive level between samples. Especially among elite athletes, previous research found that the age of peak performance occurs at about 23 and 31 years of age among male and female half-marathoners, respectively [34]. After this peak, a decrease in performance is observed, especially due to physiological changes, decrease in daily physical activity and training habits, and decline in cardiorespiratory fitness [7,35,36].

Regarding sex differences, a broad number of studies have indicated that men present better performance than women. These differences are associated with physiological, morphological, and sociocultural factors [37,38], given that men are usually more encouraged to take part in more intensive training than women and have a different perception regarding environmental security to training. Despite the higher number of male runners compared to female ones, it has been observed that there was a proportionally higher increase in the number of women who took part in marathon events between 2008–2014 (an increase of 21.61% among men; and an increase of 33.35% among women) [39], meaning that in the future, it is possible that more information related to women’s performance could be available, providing new insights regarding sexes’ differences in running performance.

In this study, we decided to present data in using three different strategies: by age interval, by running distances, and by age. In all the cases, information was stratified by sex. This decision was made to provide both runners and coaches with different possibilities to choose which information they would use for reference. For example, in both Brazilian and international races, non-elite runners are usually classified by their age group. As such, the results could allow these athletes to observe their overall running progress compared to other runners in their age cohort (e.g., São Silvestre; São Paulo Int’l marathon; Rio Marathon; Berlin marathon, New York city marathon [24,25,40]). This would potentially have the added benefit on a personal level to gauge their performance against their peers, which could further promote health and quality of life with the goal of increasing physical activity and not focusing on competition and/or place results [41]. Furthermore, participation in mass running events has been suggested to function as a form of socio-psychological therapy for those who take part [40]. Accordingly, road running fosters a sense of social togetherness, by helping to establish and maintain emotional and social ties with others during a common goal [42].

Further, the results stratified by running distances could be used to identify the general scenario observed for each distance, and this could help athletes to understand and better interpret their results, or even to organize their goals when taking part in races with different distances. Indeed, it is expected that the pace of a runner when completing a 5 km race is different from their pace in a complete 10 km race, for example [32]. Yet, it is interesting to note that the pace values for the largest distances (21 km and 42 km) at the 50th percentile for both sexes were not the highest ones, and this could be related to the fact that runners who compete in theses distances could be those who are more involved in their practice, running more regularly, with more experience, which could in part explain some of the variance in performance.

Despite all the efforts, this study is not without limitations. Firstly, we highlight the fact that we were not able to build the charts for each of the running distances, given the absence of enough information, per distance. Secondly, the sample distribution was not equivalent across the Brazilian states, or even between age categories, which limit the generalization of the results. Furthermore, notwithstanding the previous validation of the questionnaire and the large use of online surveys in research [22,23,43], it is known that self-reported information is prone to misleading data, and because of this, care must be taken in the generalization of the results. However, to the best of our knowledge, this is the first study focusing on producing percentile values for running performance specifically in a Brazilian context. Furthermore, when results found were compared against the pace of the best runners from different races performed around the country [44], it was observed that the median values in each distance race were above those observed among the best national runners. This may highlight that our results seem to be more similar real world running pace observed among non-professional Brazilian runners.

## 5. Conclusions

In summary, it is possible to show that there are differences in the running pace between sexes, in all ages, and that the running performance tends to decrease with increasing ages (given that the increment in running pace means decreases in performance). The available information is indicated to be used by runners to better understand “how well” they are compared to their peers of the same age cohort. Finally, it can used by coaches in the development training programs for non-elite and recreational runners.

## Figures and Tables

**Figure 1 healthcare-09-00829-f001:**
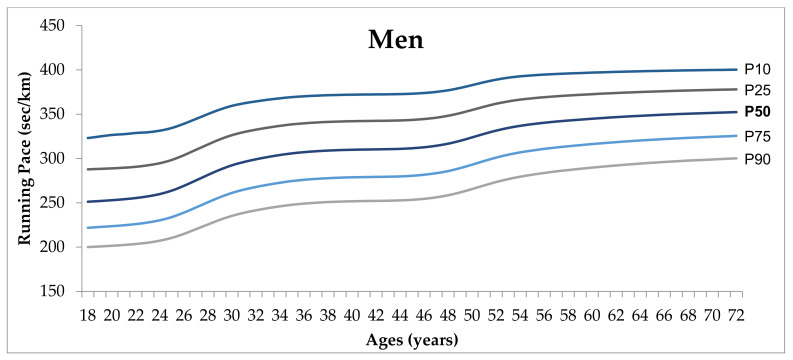
Percentile graph representing P10, P25, P50, P75, P90, of running pace for men, aged 18–72 years.

**Figure 2 healthcare-09-00829-f002:**
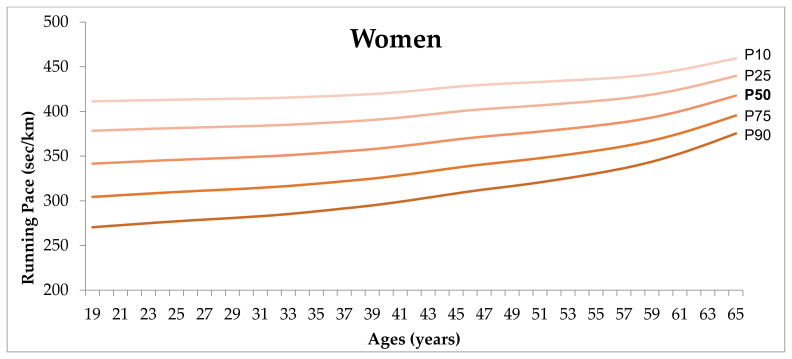
Percentile graph representing P10, P25, P50, P75, P90, of running pace for women, aged 19–65 years.

**Table 1 healthcare-09-00829-t001:** Brazilian runners’ descriptive characteristics, by total sample and stratified by sex.

Variables	Total Sample (*n* = 1152)	Female Runners (*n =* 440)	Male Runners (*n =* 712)
	Mean (SD)		
Age (years)	37.9 (9.4)	37.9 (8.5)	37.8 (9.9)
Height (cm)	170.5 (9.0)	162.9 (6.2)	175.2 (7.1)
Body mass (kg)	70.9 (12.5)	62.4 (8.8)	76.0 (11.6)
BMI (kg.m^−2^)	24.2 (3.1)	23.5 (2.9)	24.7 (3.1)
Running pace (sec)	324.1 (57.7)	358.0 (49.7)	303.2 (52.1)
Volume/week (Km)	35.4 (29.59)	26.9 (16.3)	40.6 (34.2)
	Frequency (%)		
Frequency/week			
Until 3 train	678 (58.9%)	302 (68.6%)	377 (52.9%)
>3 train	473 (41.1%)	138 (31.4%)	335 (47.1%)
Preferred distance			
5 km	300 (26.0%)	134 (31.5%)	166 (24.6%)
10 km	377 (32.7%)	161 (37.8%)	213 (32.0%)
Half-marathon	338 (29.3%)	109 (25.6%)	229 (33.9%)
Marathon	86 (7.5%)	22 (5.2%)	64 (9.5%)
Missing	51 (4.4%)	14 (3.2%)	37 (5.2%)

**Table 2 healthcare-09-00829-t002:** Running pace (seconds/km) percentiles, by age-groups and sex.

	Man			Woman			
Age Groups	P25	P50	P75	Age Groups	P25	P50	P75
Until 19 years (4)	226.25	237.50	308.75	Until 19 years (0)	-	-	-
20–24 (32)	235.00	255.00	277.50	20–24 (10)	282.50	336.50	386.25
25–29 (69)	232.50	270.00	305.00	25–29 (40)	320.00	360.00	387.50
30–34 (80)	270.00	300.00	330.00	30–34 (73)	300.00	340.00	380.00
35–39 (99)	280.00	307.00	350.00	35–39 (65)	330.00	360.00	415.00
40–44 (82)	295.00	310.00	340.00	40–44 (60)	332.50	360.00	398.75
45–49 (50)	267.75	300.00	330.00	45–49 (35)	340.00	360.00	420.00
50–54 (47)	300.00	340.00	360.00	50–54 (13)	337.50	360.00	407.50
55–59 (26)	300.00	330.00	360.00	55–59 (7)	340.00	360.00	390.00
60- 64 (5)	340.00	340.00	365.00	60–64 (1)	360.00	360.00	360.00
65–69 (57)	300.00	350.00	-	65–69 (53)	420.00	420.00	420.00
>70 years (161)	260.00	300.00	339.75	>70 years (83)	330.00	360.00	400.00

-: no available information; values in ( ) indicates the number of subjects, in each age group.

**Table 3 healthcare-09-00829-t003:** Running pace (sec/km) percentiles by sex and running distance.

	Man				Woman		
	P25	P50	P75		P25	P50	P75
5 km (166)	240.00	280.00	339.25	5 km (134)	320.00	360.00	420.00
10 km (216)	280.00	308.00	340.00	10 km (161)	340.00	360.00	405.00
Half-marathon (229)	280.00	300.00	340.00	Half-marathon (109)	320.00	340.00	360.00
Marathon (64)	265.00	300.00	330.00	42 km (22)	300.00	337.50	352.50

## Data Availability

The data are not publicly available due to ethical concerns.

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
