# Peer review of "Running Pace Percentile Values for Brazilian Non-Professional Road Runners"

_healthcare, 2021, doi:10.3390/healthcare9070829_

Round 1
Reviewer 1 Report
The authors describe that sex-specific reference curves and values for running pace for Brazilian non-professional runners using the data of 1152 participants. This manuscript is very interesting for this reviewer. However, there must be several major issues in this study. Please consider those below:
- Although this study looks very interesting for sports fields, this reviewer thinks that this manuscript does not fit the aim & scope of Healthcare journal. If the authors still want to publish in this journal, the authors should rewrite especially the Introduction and Discussion sections for Healthcare readers. How does this study contribute to medicine and health care?
- As the authors entitled "reference value," the data represented at each age group and each gender seems to be good. However, the sample size for each age group was not shown in this manuscript. This is very important information if the authors want to emphasize the "reference value." This reviewer thinks that there must be some skewness for the sample size over age groups considering the standard deviation in Table 1.
- In the Limitation section, the authors should mention that the results of this study are based on the "self-reported" information, and why the authors decided to use this methodology.
- The authors are saying the main findings: 1. Men are faster than women in all age groups; 2. There was a sex difference in running page curves over aging. Is the 1st one novel finding? There must be a lot of reports showing that men are faster than women. If there was a sex difference in running pace, why does it happen? This reviewer highly suggests the authors to discuss those main findings.
Minor issues:
5. There are several spacing errors in this manuscript. For example, L34 "...performance is__running pace," L63 "In additionto...," L162 "...where theyobserved..."
Author Response
The authors describe that sex-specific reference curves and values for running pace for Brazilian non-professional runners using the data of 1152 participants. This manuscript is very interesting for this reviewer. However, there must be several major issues in this study. Please consider those below:
- Although this study looks very interesting for sports fields, this reviewer thinks that this manuscript does not fit the aim & scope of Healthcare journal. If the authors still want to publish in this journal, the authors should rewrite especially the Introduction and Discussion sections for Healthcare readers. How does this study contribute to medicine and health care?
Author’s answers: We appreciate the comment. However, the study was developed aiming the performance, not focusing in health indicators, despite the sample be comprised by non-professional athletes. Furthermore, the manuscript was submitted in a special issue called “sports and exercise medicine”, and we believe that it can be published in this special issue.
- As the authors entitled "reference value," the data represented at each age group and each gender seems to be good. However, the sample size for each age group was not shown in this manuscript. This is very important information if the authors want to emphasize the "reference value." This reviewer thinks that there must be some skewness for the sample size over age groups considering the standard deviation in Table 1.
Author’s answers: Thanks for the comment. We included this information on table 2.
- In the Limitation section, the authors should mention that the results of this study are based on the "self-reported" information, and why the authors decided to use this methodology.
Author’s answers: We appreciate the suggestion, and we included it in the limitation section.
- The authors are saying the main findings: 1. Men are faster than women in all age groups; 2. There was a sex difference in running page curves over aging. Is the 1st one novel finding? There must be a lot of reports showing that men are faster than women. If there was a sex difference in running pace, why does it happen? This reviewer highly suggests the authors to discuss those main findings.
Author’s answers: Thanks for the comment. The discussion was adjusted.
Minor issues:
- There are several spacing errors in this manuscript. For example, L34 "...performance is__running pace," L63 "In additionto...," L162 "...where theyobserved..."
Author’s answers: Adjusted.
Reviewer 2 Report
General comments
This manuscript aims at assessing establishing sex-specific reference curves and values for running pace for non-elite Brazilian road runners, in the context of age, sex and race distance. Despite some need for better results interpretation (read below), authors manage to fulfill sufficiently their aim.
Main findings
Male runners seem to contain their running pace age-driven decrease at beginning of their thirties and end of their forties. Differently, female runners seem contain their running pace age-driven decrease at the end of their forties. <- This results interpretation should be made more explicit.
Minor comments
(line 50) Miguel-Etayo et al. [10] (viz., missing space);
(l50÷54) please, split.
Author Response
General comments
This manuscript aims at assessing establishing sex-specific reference curves and values for running pace for non-elite Brazilian road runners, in the context of age, sex and race distance. Despite some need for better results interpretation (read below), authors manage to fulfill sufficiently their aim.
Author’s answers: We appreciate this comment, and we tried to improve results’ presentation.
Main findings
Male runners seem to contain their running pace age-driven decrease at beginning of their thirties and end of their forties. Differently, female runners seem contain their running pace age-driven decrease at the end of their forties. <- This results interpretation should be made more explicit.
Author’s answers: We appreciate this comment, and we tried to improve results’ interpretation.
Minor comments
(line 50) Miguel-Etayo et al. [10] (viz., missing space);
(l50÷54) please, split.
Author’s answers: Adjusted.
Round 2
Reviewer 1 Report
Thank you for your consideration of my comments.
Clearly, the quality of the manuscript has been improved.
However, there are still several concerns in this manuscript as below:
Introduction
Well done. This clearly demonstrates the purpose of this study.
Methods
Line 110-111: The authors are saying "with a significant level of 5%." Are there any statistical comparisons in this manuscript? As you mentioned "sex difference," there should be statistical comparisons especially for Tables 2 and 3.
Results
Line 118: You are missing "s" for "1152 runner."
Line 120: As mentioned in the previous comment, there are still spacing errors, "sessions/_week." Please confirm those again over the manuscript.
Table 2: The authors show very few numbers of subjects for age groups (e.g. man and woman until 19 years). As referring to this, I am not sure that the authors can say "reference values" for those age groups. The authors have to explain whether the data size is good enough for saying "reference values" in the Discussion section.
Figures 1 and 2: The font type of words in these Figures is not the same as in texts. Please unify those.
Discussion
Showing what are the main findings in this study in the first paragraph would be easier for readers to understand. Then, the authors should discuss those in order. For example, if the 1st finding is a running pace curve, they should discuss it in Paragraph 2.
Line 156-167: I am not sure whether this paragraph is necessary for this manuscript. It is because the sentences here appear to be not related to the results in this study.
Line 179-185: I can see that the authors try to explain the running pace curve here. However, the authors should explain why the running pace in this study peaked at, for example, the beginning of their 30's in men in this study based on citation 33. For women as well.
Line 185-188: The authors try to explain the 3rd finding in this study. However, do the authors still think that the 3rd finding, "men are faster than women," is one of the main findings in this study? To this reviewer, as the authors discuss the following paragraphs (Line 189-203; Line 204-213), information on sex and running distance would be more important findings for readers as "reference values." Please re-organize those paragraphs.
Line 199: There may be the spacing error, "...not focusing on _ competition..."
References
There are a lot of mistakes here. For example, several references do not have journal names (e.g. references 6, 12, 14, and so on). For reference 15, the DOI number is probably wrong. Please confirm again carefully.
Author Response
Thank you for your consideration of my comments.
Clearly, the quality of the manuscript has been improved.
However, there are still several concerns in this manuscript as below:
Introduction
Well done. This clearly demonstrates the purpose of this study.
Methods
Line 110-111: The authors are saying "with a significant level of 5%." Are there any statistical comparisons in this manuscript? As you mentioned "sex difference," there should be statistical comparisons especially for Tables 2 and 3.
Authors answers: We appreciate this comment. We included differences regarding distance of preference between sexes.
Results
Line 118: You are missing "s" for "1152 runner."
Authors answers: Adjusted
Line 120: As mentioned in the previous comment, there are still spacing errors, "sessions/_week." Please confirm those again over the manuscript.
Authors answers: We revised the manuscript, and tried to adjust.
Table 2: The authors show very few numbers of subjects for age groups (e.g. man and woman until 19 years). As referring to this, I am not sure that the authors can say "reference values" for those age groups. The authors have to explain whether the data size is good enough for saying "reference values" in the Discussion section.
Authors answers: Thanks for this comment. We adjusted and changed “reference values” to “percentile values”.
Figures 1 and 2: The font type of words in these Figures is not the same as in texts. Please unify those.
Authors answers: Adjusted
Discussion
Showing what are the main findings in this study in the first paragraph would be easier for readers to understand. Then, the authors should discuss those in order. For example, if the 1st finding is a running pace curve, they should discuss it in Paragraph 2.
Authors answers: We tried turn more clear.
Line 156-167: I am not sure whether this paragraph is necessary for this manuscript. It is because the sentences here appear to be not related to the results in this study.
Authors answers: The paragraph was removed, since the same information was presented in introduction.
Line 179-185: I can see that the authors try to explain the running pace curve here. However, the authors should explain why the running pace in this study peaked at, for example, the beginning of their 30's in men in this study based on citation 33. For women as well.
Authors answers: We tried to improve the explanation.
Line 185-188: The authors try to explain the 3rd finding in this study. However, do the authors still think that the 3rd finding, "men are faster than women," is one of the main findings in this study? To this reviewer, as the authors discuss the following paragraphs (Line 189-203; Line 204-213), information on sex and running distance would be more important findings for readers as "reference values." Please re-organize those paragraphs.
Authors answers: Adjusted.
Line 199: There may be the spacing error, "...not focusing on _ competition..."
Authors answers: Adjusted
References
There are a lot of mistakes here. For example, several references do not have journal names (e.g. references 6, 12, 14, and so on). For reference 15, the DOI number is probably wrong. Please confirm again carefully.
Authors answers: Adjusted